# Neuregulin 4 Downregulation Alters Mitochondrial Morphology and Induces Oxidative Stress in 3T3-L1 Adipocytes

**DOI:** 10.3390/ijms252111718

**Published:** 2024-10-31

**Authors:** Francisco Díaz-Sáez, Cristina Balcells, Laura Rosselló, Iliana López-Soldado, Montserrat Romero, David Sebastián, Francisco Javier López-Soriano, Sílvia Busquets, Marta Cascante, Wifredo Ricart, José Manuel Fernández-Real, José María Moreno-Navarrete, Julián Aragonés, Xavier Testar, Marta Camps, Antonio Zorzano, Anna Gumà

**Affiliations:** 1Departament de Bioquímica i Biomedicina Molecular, Facultat de Biologia, Universitat de Barcelona (UB), Av. Diagonal 643, 08028 Barcelona, Spain; frandiazsaez@gmail.com (F.D.-S.); crisgatsu@gmail.com (C.B.); laura.ross.rull@gmail.com (L.R.); ilopez-soldado@ub.edu (I.L.-S.); montserrat.romero@ub.edu (M.R.); flopez@ub.edu (F.J.L.-S.); silviabusquets@ub.edu (S.B.); martacascante@ub.edu (M.C.); xtestar@ub.edu (X.T.); martacamps@ub.edu (M.C.); 2Institut de Biomedicina de la Universitat de Barcelona (IBUB), Universitat de Barcelona (UB), 08028 Barcelona, Spain; 3CIBER de Diabetes y Enfermedades Metabólicas Asociadas, Instituto de Salud Carlos III, 28029 Madrid, Spain; dsebastian@ub.edu; 4Institute for Research in Biomedicine (IRB Barcelona), The Barcelona Institute of Science and Technology (BIST), 08028 Barcelona, Spain; 5Departament de Bioquímica i Fisiologia, Facultat de Farmàcia i Ciències de l’Alimentació, Universitat de Barcelona, Av. Joan XXIII, 27-31, 08028 Barcelona, Spain; 6Centro de Investigación Biomédica en Red (CIBER) de *Enfermedades Hepáticas y Digestivas*, Instituto de Salud Carlos III, 28029 Madrid, Spain; 7Department of Medical Sciences, University of Girona, Carrer Emili Grahit, 77, 17003 Girona, Spain; wricart@idibgi.org (W.R.); jmfreal@idibgi.org (J.M.F.-R.); jmoreno@idibgi.org (J.M.M.-N.); 8Department of Diabetes, Endocrinology and Nutrition, Girona Biomedical Research Institute (IDIBGI), Carrer del Dr. Castany, s/n, 17190 Salt, Spain; 9Centro de Investigación Biomédica en Red (CIBER) de *Fisiopatología de la Obesidad y Nutrición* (CB06/03/010), Instituto de Salud Carlos III, 28029 Madrid, Spain; 10Research Unit, Hospital of Santa Cristina, Research Institute Princesa, University Hospital of la Princesa, Autonomous University of Madrid, c/Maestro Vives, 2, 28009 Madrid, Spain; julian.aragones@salud.madrid.org; 11Centro de Investigación Biomédica en Red (CIBER) de *Enfermedades Cardiovasculares*, Instituto de Salud Carlos III, 28029 Madrid, Spain

**Keywords:** Neuregulin 4, mitochondria, oxidative stress, 3T3-L1 adipocytes, inflammation, ErbB4, insulin resistance

## Abstract

Neuregulin 4 (Nrg4) is an adipokine that belongs to the epidermal growth factor family and binds to ErbB4 tyrosine kinase receptors. In 3T3-L1 adipocytes, the downregulation of *Nrg4* expression enhances inflammation and autophagy, resulting in insulin resistance. Here, we searched for the causes of this phenotype. Nrg4 knockdown (Nrg4 KD) adipocytes showed a significant reduction in mitochondrial content and elongation, along with a lower content of the mitochondria fusion protein mitofusin 2 (MFN2), and increased H_2_O_2_ production compared to the control scrambled cells (Scr). The antioxidant N-acetylcysteine reversed the oxidative stress and reduced the gene expression of the pro-inflammatory cytokine tumor necrosis factor α (TNFα). Nrg4 KD adipocytes showed enhanced lipolysis and reduced lipogenesis, in addition to a significant reduction in several intermediates of the Krebs cycle. In summary, Nrg4 downregulation in adipocytes affects mitochondrial content and functioning, causing impaired cellular metabolism, which in turn results in oxidative stress, inflammation, and insulin resistance.

## 1. Introduction

Adipose tissues play a central role in the regulation of whole-body basal metabolic homeostasis, functioning as endocrine tissues through the release of cytokines with local and distal effects [1,2]. Neuregulin 4 (Nrg4) is an adipokine that is highly expressed in brown adipose tissue (BAT) but also in white adipose tissue (WAT) that protects against insulin resistance [3,4]. Obesity, a condition characterized by chronic low-grade inflammation, negatively impacts the capacity of brown and white adipocytes to express Nrg4 in humans [5]. In this regard, it was shown that the pro-inflammatory cytokine tumor necrosis factor α (TNFα) disrupts *Nrg4* expression in 3T3-L1 adipocytes [5,6,7]. Interestingly, Nrg4 behaves as an anti-inflammatory factor. It prevents macrophages’ polarization to a pro-inflammatory profile [8] and reduces the expression of inflammatory cytokines in several tissues, among them WAT [5,6,9,10]. In addition to its anti-inflammatory role, Nrg4 enhances angiogenesis [11] and induces sympathetic innervation of WAT upon cold exposure [12]. Indeed, Nrg4 was proposed as a marker of beige adipocytes [13]. Previously, we described that 3T3-L1 adipocytes’ knockdown for *Nrg4* expression (Nrg4 KD) results in dramatic insulin resistance, which is caused by a high expression of pro-inflammatory cytokines, leading to a reduction in insulin receptor expression [7]. In addition, these adipocytes show an enhanced autophagic flux, which leads to an increase in the degradation of the intracellular vesicles that contain the insulin-sensitive glucose transporter GLUT4 [7]. In type 2 diabetes, insulin resistance is characterized by inflammation and altered autophagy in all insulin-targeted tissues, and these factors are closely interconnected with mitochondrial dysfunction and metabolic cellular stress [14]. In this study, we searched for potential drivers of cell-autonomous inflammation and dysregulated autophagy associated with insulin resistance in Nrg4 KD adipocytes. Here, we observed that Nrg4 deficiency in adipocytes alters mitochondria and unbalances cellular energy metabolism, entailing harmful consequences that drive insulin resistance.

## 2. Results

### 2.1. Adipocytes Silenced for the Expression of the Adipokine Neuregulin 4 Showed a Lower Mitochondrial Content

In a previous study [7], we generated a stable Nrg4 KD cell line in 3T3-L1 adipocytes, based on the use of lentiviral vectors expressing a shRNA for *Nrg4*, or a scrambled sequence for the control cell line. As mentioned, Nrg4 KD adipocytes showed marked insulin resistance characterized by enhanced expression of inflammatory cytokines and increased autophagic flux [7]. Given that these two alterations could arise from mitochondria dysregulation [15], we examined mitochondrial proteins’ expression (Figure 1a,b) in the stable cell line silenced for the *Nrg4* expression (Appendix A). No altered expression was observed for several oxidative phosphorylation (OXPHOS) complex subunits, including Ndufv1 of Complex I, Uqcrh of Complex III, and Atp5l of Complex V. This was also confirmed at the protein level for the 39 KD subunit of Complex I. Mitochondrial biogenesis did not seem to be affected since no change in the expression of peroxisome proliferator-activated receptor γ co-activator 1α, PGC1α, was detected (Figure 1b). Under these conditions, we observed a significant decrease in the protein content of the inner mitochondrial membrane TIM44. Given these results, we analyzed other reliable markers of mitochondrial content, such as the mitochondrial DNA copy number, which showed a significant decrease in Nrg4 KD adipocytes (Figure 1c). This finding was further confirmed by analysis of the mitochondrial content with MitoTracker^TM^ green-staining (Figure 1d). Therefore, Nrg4 KD adipocytes showed lower mitochondrial content than control cells.

### 2.2. Mitochondria Fragmentation in Neuregulin 4 Knockdown Adipocytes

Next, we examined whether the morphology of the mitochondrial network was affected in Nrg4 KD adipocytes. Initially, we determined the expression of proteins involved in mitochondrial dynamics, specifically those required for membranes’ fusion, namely, the inner mitochondria membrane protein optic atrophy-1 (OPA1) and the outer mitochondria membrane protein mitofusin 2 (MFN2) [16]. Our data (Figure 2a,b) indicated no alteration of *Opa1* and *Mfn2* expression or protein content of OPA1. In contrast, the protein content of MFN2 was significantly decreased in these adipocytes. A confocal microscope analysis of the mitochondrial network, using the MitoTracker^TM^ red probe (Figure 2c,d), showed a marked loss of elongated mitochondria, which formed condensed structures in Nrg4 KD adipocytes. To determine whether the enhanced autophagy present in Nrg4 KD adipocytes [7] was relevant for the reduced abundance of the mitochondria proteins TIM44 and MFN2, we blocked autophagy with bafilomycin A1, a disruptor of lysosomal function. The protein content of both TIM44 and MFN2 in these adipocytes was restored upon treatment with bafilomycin A1 (Figure 2e), suggesting that enhanced autophagy affects the content of these mitochondrial proteins.

### 2.3. Neuregulin 4 Knockdown Adipocytes Showed Evidence of Oxidative Stress

Next, we examined whether Nrg4 KD adipocytes also showed oxidative stress. Analysis of the H_2_O_2_ levels indicated a significant increase in these adipocytes when compared to Scr cells (Figure 3a). Accordingly, there was an increase in the expression of superoxide dismutase 2 (*Sod2*), which senses and counteracts the enhanced oxidative stress [17] (Figure 3b). Although Nrg4 KD adipocytes showed increased expression of the *uncoupler protein 1* (*Ucp1*) gene, the UCP1 protein content was significantly reduced. Under these conditions, the mitochondrial membrane potential was higher in Nrg4 KD adipocytes (Figure 3b–d). Given that these adipocytes overexpress TNF*α* [7], we studied whether this pro-inflammatory cytokine alters UCP1 protein content. To this end, we treated control and Nrg4 KD adipocytes with 10 ng/mL TNFα for 24 h. This treatment had no effect on UCP1 in Scr cells, although it significantly contributed to the UCP1 decrease in KD adipocytes (Figure 3d). To determine whether the pro-inflammatory scenario in Nrg4 KD adipocytes was due to the manifestation of oxidative stress, we treated Scr and Nrg4 KD adipocytes with the antioxidant compound N-acetylcysteine (NAC) 1 mM for 24 h, which restored the superoxide H_2_O_2_ to control levels in KD adipocytes (Figure 3e). In this condition, the expression of *Tnfα* was totally abrogated, thereby indicating that oxidative stress could lead to the overexpression of this pro-inflammatory cytokine (Figure 3f). In this regard, NAC caused a significant increase in IκB, the natural inhibitor of the transcription factor NFκB, which induced the expression of the pro-inflammatory cytokines (Figure 3g). Moreover, NAC contributed to partially restoring the content of insulin receptor (Figure 3g) and insulin-sensitive glucose transporter GLUT4 (Appendix A) in Nrg4 KD adipocytes. Taken together, our data suggest that altered mitochondrial functioning in Nrg4 KD adipocytes causes oxidative stress which, in turn, enhances the expression of pro-inflammatory cytokines that negatively impact insulin effectors. 

### 2.4. Altered Lipid and Glucose Metabolism in Neuregulin 4 Knockdown Adipocytes

Oxidative stress and inflammation lead to metabolic dysfunction in adipocytes [18]. Therefore, lipid storage and metabolism were initially analyzed in Nrg4 KD adipocytes (Figure 4). Triacylglyceride (TAG) pools were significantly lower in differentiated Nrg4 KD adipocytes (Figure 4a). Cells treated with conditioned media from the control Scr adipocytes (M), containing endogenously expressed Nrg4, or with human recombinant neuregulin 4 (hrNRG4) led to normalized TAG pools (Figure 4b). The lower levels of TAG observed in Nrg4 KD adipocytes could be the result of higher lipolysis and/or lower lipogenesis rates in KD adipocytes. We examined lipolysis, measured as the adipocytes’ production of glycerol released to the media over the last 24 h of differentiation, while lipogenesis was determined by incubating cells with ^14^C-glucose for 90 min and measuring the radioactivity incorporated into the lipid fraction. Both lipolysis and lipogenesis were significantly altered, with the lipolysis rate being higher and lipogenesis being lower in Nrg4 KD than in Scr adipocytes (Figure 4c,d). As expected, insulin-stimulated lipogenesis was completely blocked in Nrg4 KD adipocytes (Figure 4d), in keeping with previous data on insulin resistance in this model [7]. Future studies should address whether lipophagy is enhanced in Nrg4 KD adipocytes, as suggested by the higher autophagic flux previously described in these cells. An analysis of glucose metabolism, by determining lactate release to the medium from D6 to D7 of differentiation but also glucose oxidation at D7 for 3 h using ^14^C-glucose and measuring radiolabeled CO_2_ production, revealed a significant decrease in lactate production while no difference was observed in the capacity of Nrg4 KD adipocytes to oxidize glucose (Figure 4e,f). Again, insulin-stimulated glucose oxidation was abrogated in KD adipocytes (Figure 4f). Overall, the decrease in TAG content observed in Nrg4 KD adipocytes appeared to depend on increased lipolysis and reduced conversion of glucose to lipids. Instead, glucose seemed to be primarily directed to oxidation in these adipocytes.

### 2.5. Classical Lipolysis in Differentiated Neuregulin 4 Knockdown Adipocytes

To discern the cause of enhanced TAG degradation in Nrg4 KD adipocytes, we examined the expression of lipases involved in the classical lipolysis pathway. Unexpectedly, the expression of lipases was markedly reduced in these cells (Figure 5a,b). Nevertheless, the significant decrease in the content of perilipin-1 protein, which is involved in the activation of adipose triacylglyceride lipase (ATGL) [19], and the increase in the phosphorylation of hormone-sensitive lipase (HSL), which enhances HSL activity, suggest that the remaining lipases were more activated in Nrg4 KD adipocytes (Figure 5b). Further studies are needed to decipher whether the higher autophagy also contributes to the TAG degradation in these cells. 

### 2.6. Alteration in the Content of Polar Metabolites Associated with Oxidative Metabolism in Neuregulin 4 Knockdown Adipocytes

Next, we examined the abundance of polar metabolites involved in oxidative metabolism whose alterations may explain the adaptation of adipocytes to Nrg4 deficiency. Cellular lysates obtained from Scr and Nrg4 KD adipocytes at day 7 of differentiation were analyzed by gas chromatography-mass spectrometry (GC-MS). Nrg4 KD cells showed a marked decrease in the intracellular content of the Krebs cycle intermediates’ citrate, fumarate, and malate compared to Scr adipocytes (Figure 6a). Interestingly, an analysis of the cellular conditioned media obtained from day 6 to day 7 of differentiation, used to calculate the consumption and production rates of some of these metabolites (Figure 6b), revealed that while citrate was secreted by Nrg4 KD adipocytes, these cells were taking up succinate from the medium, thereby preserving the intracellular content of this intermediate. In terms of the Krebs cycle functionality, the loss of citrate may have been compensated by the uptake of succinate. Of note, amino acids involved in anaplerotic reactions of the Krebs cycle also decreased in Nrg4 KD cells when compared with Scr counterparts. That was the case of glutamine and proline, which render α-ketoglutarate, and of phenylalanine and methionine, which render fumarate and succinyl CoA, respectively. The intracellular content of glycine also decreased significantly in Nrg4 KD adipocytes. This amino acid together with glutamate and cysteine, which also tended to decrease in Nrg4 KD adipocytes, may contribute to sustaining the intracellular levels of glutathione, whose content would be essential for adaptation to oxidative stress. The decrease in malate could result from the increased activity of the malic enzyme to generate the redox potential necessary to preserve glutathione under reducing conditions and, in turn, contribute to maintaining pyruvate levels in Nrg4 KD adipocytes. In addition, low malate levels could promote the main anaplerotic reaction of the Krebs cycle, which involves the carboxylation of pyruvate to oxaloacetate. This, in turn, would cause a decrease in the conversion of pyruvate to lactate in Nrg4 KD adipocytes, which was observed by GC-MS and confirmed the results described above. Nrg4 KD adipocytes showed a marked increase in the branched-chain amino acids (BCAA) isoleucine and leucine (Figure 6a). Increases in circulating BCAA were associated with insulin resistance both in humans and rodents [20]. In this regard, it was postulated that the impairment of BCAA catabolism, a process that occurs in mitochondria, causes the accumulation of non-metabolized intermediates, such as their α-keto acids, which interfere with insulin signaling and its metabolic effects [21,22]. Therefore, the altered mitochondrial function observed in Nrg4 KD adipocytes could affect BCAA metabolism. Overall, the mitochondrial disturbances observed in Nrg4 KD adipocytes appear to be closely related to oxidative stress and insulin resistance.

## 3. Discussion

Previous studies reported an inverse association between adipose tissue expression of Nrg4 and the circulating levels of this protein with insulin resistance in humans and in mouse models of obesity and type 2 diabetes (reviewed in [23]). Here, we provided further evidence of the relevant role of Nrg4 as a protective adipokine, and we demonstrated that its deficiency alters mitochondria morphology and metabolism, drives cell accumulation of BCAA, and induces oxidative stress in adipocytes. Taken together, this pattern of changes is characteristic of insulin resistance, along with the cell-autonomous inflammation previously described in this cell model [7]. However, Nrg4 KD adipocytes maintain their capacity to differentiate and, to some extent, compensate for these metabolic alterations without critical consequences. This compensation could be due to two factors. First, the adipocyte model characterized here preserves certain levels of the Nrg4 expression, which may be enough to preserve cell viability. Second, Nrg4 expression occurs late in adipogenesis [7] and such partially differentiated adipocytes may have the capacity to adapt to the low levels of Nrg4, although with an impact on metabolic homeostasis. 

There is considerable evidence indicating mitochondrial alterations in Nrg4 KD adipocytes. One such alteration is the decrease in the mitochondrial content, which may be more related to the degradation of damaged mitochondria than to a defect in mitochondrial biosynthesis, as reflected by the unaltered expression of the co-activator PGC1α and the recovery in the content of the mitochondrial membrane proteins MFN2 and TIM44 upon autophagy inhibition with bafilomycin A1. Intriguingly, the content of other mitochondrial proteins, such as the OXPHOS subunits and the mitochondria membranes’ fusion protein OPA1, was unaltered in Nrg4 KD adipocytes, suggesting a selective mitochondrial quality control process. Particularly interesting was the loss of the mitochondrial outer membrane fusion protein MFN2. In addition to participating in mitochondria elongation/fragmentation dynamics, MFN2 is involved in the tethering of mitochondria to other organelles, including the endoplasmic reticulum (ER), lipid droplets, and lysosomes [24,25,26], and its deficiency could affect cell homeostasis. MFN2 deficiency was associated with the development of ER stress, elevated cellular content of hydrogen peroxide, and insulin resistance [27,28]. Interestingly, there is a reciprocal relationship between mitochondrial fragmentation and oxidative stress, since cell treatment with hydrogen peroxidase enhances mitochondrial fragmentation as a result of changes in fission and fusion rates of the mitochondria membranes [29]. This relationship hides the causal effect of the oxidative stress observed in the Nrg4 KD adipocytes, although it is tempting to propose that the decrease in MFN2 leads to mitochondrial and ER alterations resulting in an increased formation of superoxides. In obesity, mitochondrial fragmentation is associated with a decrease in MFN2 in skeletal muscle [30]. Mitochondrial elongation/fragmentation was postulated to occur as a cellular adaptive response to energy demands. [31]. Thus, while starvation involves mitochondrial elongation and a higher efficiency in ATP synthesis, the excess of nutrients leads to mitochondrial fragmentation, in which respiration is uncoupled. This was not the case for Nrg4 KD adipocytes in which mitochondria fusion arrest occurred in parallel with a decrease in UCP1 protein content, regardless of the increase in *Ucp1* gene expression. These observations indicate that such fragmented mitochondria are less uncoupled, which is confirmed by the increase in the mitochondrial membrane potential. Therefore, Nrg4 KD adipocytes emerge as a cellular model in which mitochondrial fragmentation, probably due to the higher expression of pro-inflammatory cytokines such as TNFα [32], coexists with a harmful oxidative stress that the cells should reduce. This may explain the decrease in UCP1. In this regard, our data enforce this notion that involves TNFα since the treatment of Nrg4 KD adipocytes with this cytokine further reduces UCP1, but this does not happen in Scr adipocytes. 

We observed the activation of lipolysis in Nrg4 KD adipocytes, which is parallel to increased *Tnfα* expression [33]. Therefore, it is expected that a substantial proportion of the free fatty acids generated undergo oxidation. In this regard, when adapted to an excess of nutrients, mitochondrial fragmentation enhances lipid consumption as a source of energy by reducing malonyl CoA inhibition of the carnitine palmitoyl transport system 1 (CPT1) [34]. It remains unknown whether Nrg4 KD adipocytes oxidize more free fatty acids than glucose, but we observed that glucose oxidation was maintained at the same rate as control cells. The connection between lipid droplets and mitochondria seems to play a relevant role in delivering fatty acids to be oxidized. It was proposed that this link between lipid droplets and mitochondria is mediated by mitochondrial MFN2, which binds to perilipin-1 in the lipid droplets [35]. Nrg4 KD adipocytes may have difficulties channeling free fatty acids to the mitochondria since they not only showed a decrease in MFN2 but also in perilipin-1. This observation suggests that the channeling of free fatty acids from lipid droplets to mitochondria limits fatty acid oxidation, which may explain the sustained basal rates of glucose oxidation. However, lactate production decreased in Nrg4 KD adipocytes. Again, the clue may be the occurrence of oxidative stress in these cells. To protect the cells against oxidative stress, glucose metabolism could permit the preservation of NAPH+H^+^ content to conserve reduced glutathione. In adipocytes, two pathways, namely, the pentose phosphate pathway and the malic enzyme, must ensure this redox potential. The lower levels of lactate produced by Nrg4 KD adipocytes suggest that part of the glucose was oxidized by the pentose phosphate pathway. Additionally, the decrease in different intermediates of the Krebs cycle in Nrg4 KD adipocytes, among them the malate, suggests that it is greatly metabolized by the malic enzyme generating pyruvate and NAPH+H^+^. Pyruvate could be recycled to generate oxaloacetate by the pyruvate carboxylase, which in turn would be a source of malate by the malate dehydrogenase, especially in conditions in which fumarate concentrations are lower, as observed in Nrg4 KD adipocytes. The α-ketoglutarate dehydrogenase (α-KGDH) is highly sensitive to reactive oxygen species (ROS) and the inhibition of this enzyme could be critical in the metabolic deficiency induced by oxidative stress [36]. This may explain why Nrg4 KD adipocytes uptake succinate from the media and increase the use of amino acids that contribute anaplerotically to sustaining the Krebs cycle rate. An increase in succinate consumption may also contribute to superoxides’ formation in Nrg4 KD adipocytes. It was postulated that succinate plays several roles as a signaling molecule (reviewed in [37,38]). One of them is the induction of pro-inflammatory cytokines’ expression [39], but succinate may also be involved in extensive ROS generation by reverse electron transport at mitochondrial Complex I [40]. 

Adipocytes display a “broken” Krebs cycle, exporting mitochondrial citrate to the cytosol as a vehicle to deliver acetyl CoA for lipogenesis. Glucose utilization for lipogenesis is reduced in Nrg4 KD adipocytes, and this could be caused by several factors. First, in a cell context of oxidative stress, the higher demand for NAPH+H^+^ limits the availability of this reducing potential for fatty acid synthase (FAS). Second, citrate is catabolized by the ATP-citrate lyase (ACL) in the cytosol to form acetyl CoA. It was shown that this enzyme is regulated by phosphorylation by the same enzymatic machinery as the one that controls branched-chain ketoacid dehydrogenase (BCKDH) [41], which appears to be impaired in Nrg4 KD adipocytes, as reflected by the increase in branched-chain amino acids’ content. Deficiency in BCAA metabolization is a hallmark of insulin resistance in obesity and type 2 diabetes [42]. Third, the lower intracellular content of citrate and a presumably higher content of free fatty acids drive the low activity of acetyl CoA carboxylase that transforms acetyl CoA to malonyl CoA, which, in turn, would facilitate a greater CPT1 activity, allowing the entry of free fatty acids to mitochondria for oxidation. However, a question remains, namely, why citrate release into the medium is greater for Nrg4 KD adipocytes while its intracellular concentration is lower than that in the control cells.

Here, we provided evidence supporting the essential role of Nrg4 as a beneficial adipokine whose deficiency leads to oxidative stress as a result of impaired metabolic homeostasis, probably initiated by dysfunctional mitochondria, which brings about severe insulin resistance. Future studies should delineate the progression features that are directly regulated by Nrg4 in adipocytes and determine whether this phenotype is reproducible in other insulin-sensitive tissues, such as the liver and skeletal muscle to provide new pharmacological targets to prevent or reverse metabolic diseases.

## 4. Materials and Methods

### 4.1. Reagents

High-glucose (4.5 g/L) containing Dulbecco’s modified Eagle’s medium (DMEM #L0104-500) was purchased from Biowest (Nuaillé, France). Calf serum (#16170078), fetal bovine serum (FBS) (#10270106), penicillin/streptomycin (#15140122), EBSS media (#24010043), and puromycin (#A1113803) were purchased from Gibco (Tavarnuzze, Italy). The PureLink^TM^ HiPure Plasmid DNA Kit (#K210005), RNA Mini Kit columns (#12183018A), deoxynucleotides (#R0181), oligodTs (#18418020), RNAseOUT™ solution (#10777019), SuperScript™ II reverse transcriptase (#18064014), MitoTracker^TM^ Green FM (#M7514), and MitoTracker^TM^ Red CMXRos (#M7512) were purchased from Invitrogen (Waltham, MA, USA). SYBR^TM^ Green PCR Master Mix (#4367659) was supplied by Applied Biosystems (Waltham, MA, USA). Bioactive recombinant TNFα (#300-01A) was obtained from Peprotech (London, UK). Human recombinant neuregulin 4 (#RKQ8WWG) was purchased from Reprokine (Rehovot, Israel). Bafilomycin A1 (#sc-201550) was purchased from Santa Cruz Biotechnology (Dallas, TX, USA). Protease inhibitor cocktail (#78430) and the Pierce™ Bicinchoninic Acid Protein Assay Kit (BCA) (#23225) were purchased from Thermo Scientific (Waltham, MA, USA). Phosphatase inhibitor (#04906845001) was obtained from Roche (Basel, Switzerland) Molecular-weight size marker Hyper PAGE was obtained from Bioline (London, UK). PVDF membranes (#IPVH00010) were purchased from MERCK Millipore (Darmstadt, Germany). The anti-GLUT4 polyclonal antibody, raised against the 15 C-terminal amino acid residues, OSCRX, was produced in our laboratory [43]. The rabbit antibody anti-insulin receptor β chain (InsR) (#611277) was purchased from BD Transduction Laboratories (San Jose, CA, USA). The rabbit antibodies anti-GAPDH (#5174T), anti-ATGL (#21385), anti-pS(660)-HSL (#4126), and anti-HSL (#18381) were purchased from Cell Signaling Technologies (Danvers, MA, USA). The rabbit antibodies anti-IkB-α (#sc-203) and perilipin-1 (#Sc-67164) were obtained from Santa Cruz Biotechnology (Dallas, TX, USA). The rabbit antibodies anti-TIM44 (#Ab244466), anti-PGC1α (#Ab54481), and anti-UCP1 (#Ab10983) and the mouse antibodies anti-MFN2 (#Ab56889) and anti-OPA1 (#Ab42364) were obtained from Abcam (Cambridge, UK). The mouse antibody anti-39 KDa subunit of Complex I (#PAS39270) was obtained from Invitrogen (Waltham, MA, USA). Horseradish peroxidase (HRP)-conjugated anti-rabbit (#711-035-152) and anti-mouse (#715-035-150) secondary antibodies were obtained from Jackson ImmunoResearch (Soham, UK). The ECL^TM^ Kit (#15387655) and the D-[U-^14^C] glucose (#CFB96) were purchased from Amersham GE Healthcare (Buckinghamshire, UK). Bovine serum albumin (BSA) (#6003), D-(+)-glucose (#G8270), human recombinant insulin (#I5500), dexamethasone (#D2915), rosiglitazone (#R2408), 3-isobutyl-1-methylxanthine (IBMX) (#I5879), ampicillin (#A1593), N-acetylcysteine (NAC) (#A7250), and gene-specific primers used for quantitative-PCR analysis were obtained from Sigma-Aldrich (St. Louis, MO, USA). Other commonly used chemicals were purchased from Sigma-Aldrich. 

### 4.2. T3-L1 Adipocyte Cell Culture and Generation of the NRG4 Knockdown Line

The 3T3-L1 preadipocytes were grown and differentiated for 7 days as previously described [7]. Lentiviral viruses expressing a short-hairpin interfering RNA (shRNA) against *Nrg4* (GCCTGGTAGAGACAAACAATA) or a scrambled (Scr) control (SHC002) were used to transduce 3T3-L1 preadipocytes [7,44], obtaining stable Scr shRNA-expressing cells and Nrg4 KD preadipocytes’ clones, which were maintained in 2.5 μg/mL puromycin during the proliferative stage. 

### 4.3. Cell Treatments

To analyze insulin action, fully differentiated adipocytes were serum-starved for 16 h and supplemented with 0.2% (*w*/*v*) BSA. Insulin action in adipocytes was assessed by treating cells with 100 nM insulin for 30 min at 37 °C. For studies with TNFα, adipocytes were treated with 10 ng/mL TNFα at day 6 of differentiation and for 24 h. Rescue studies for Nrg4 silenced cells were performed by incubating the cells with 2 mL of conditioned medium from fully differentiated Scr 3T3-L1 adipocytes at day 5 of differentiation for 48 h. In parallel, Nrg4 KD cells were treated with 50 ng/mL human recombinant Nrg4 (hrNRG4) for the final 48 h of differentiation. At day 6 of differentiation, adipocytes were treated with 1 mM N-acetylcysteine (NAC) for 24 h. Finally, to assess autophagy, differentiated 3T3-L1 adipocytes were starved for 2 h with amino acids starving media (EBSS) and then treated with 200 nM bafilomycin A1 for 2 h. 

### 4.4. Determination of Mitochondria Mass, Mitochondrial Network, and the Mitochondrial Membrane Potential by Flow Cytometry and Confocal Microscopy

We used the MitoTracker^TM^ green probe to analyze mitochondrial mass via flow cytometry in living differentiated adipocytes, while the red probe was used to analyze the mitochondrial morphology by confocal microscopy in fixed adipocytes, as previously described [45]. Adipocytes were incubated for 30 min at 37 °C with 100 nM MitoTracker^TM^ red probe and were fixed with 3% (*w*/*v*) paraformaldehyde. Coverslips were mounted with the FluoromountTM mounting medium. Images were taken in an LSM 880 confocal microscope from Zeiss and using the Zen Black and Blue software from Zeiss. Regarding MitoTracker^TM^ green-staining, adipocytes were differentiated and treated with 100 nM MitoTracker^TM^ green probe for 20 min. Cells were detached and resuspended in pre-warmed KRHB buffer in flow cytometry tubes to be analyzed in a Gallios Flow Cytometer from Beckman Coulter Inc. To study the mitochondrial membrane potential of differentiated adipocytes, the MitoProbe^TM^ TMRM Assay Kit (#Ab228569) from Thermo Scientific was used, following the manufacturer’s instructions. To this end, adipocytes were grown and differentiated in 6-well plates, and at day 7, the cells were supplemented with 50 nM TMRM for 30 min in the dark at 37 °C. Next, adipocytes were collected, resuspended in 500 μL of pre-warmed KRP-HEPES buffer in flow cytometry tubes, and analyzed in a Gallios Flow Cytometer. 

### 4.5. Quantification of Mitochondrial DNA 

Mitochondrial DNA (mtDNA) copy number was determined, as previously described [44,45]. Mitochondrial and nuclear DNA was isolated and amplified with specific oligodeoxynucleotides to calculate the mitochondrial copy number per cell.

### 4.6. Measurement of Glucose Oxidation and De Novo Lipogenesis

Glucose oxidation was determined in serum-starved control and 100 nM insulin-treated adipocytes for 30 min, using radiolabeled D-[U-^14^C] glucose (#CFB96, Amersham) and analyzing the captured radioactive CO_2_, on a β-counter Tri-Carb 2100TR from Packard Instrument Company, as previously described [46]. Data were normalized to protein content in cell lysates. Protein concentration was quantified with a Pierce™ BCA Protein Assay Kit (#23225, Thermo Scientific, Waltham, MA, USA). We also measured de novo lipogenesis, in control and insulin-treated adipocytes, using radiolabeled D-[U-^14^C] glucose, 0.5 μCi/μL for 90 min, followed by the lipid extraction with DOLE reagent, containing 40 mL of isopropanol, 10 mL of heptane, and 1 mL of H_2_SO_4_; the separation of the organic lipid containing fraction from the aqueous fraction and determination of the radioactivity in the former one at the β-counter was conducted.

### 4.7. Spectrophotometry and Fluorimetry Determination of Different Metabolites

The triacylglycerol (TAG) content of adipocytes was determined in total cell lysates using an Enzymatic-spectrophotometric Glycerol Phosphate Oxidase/Peroxidase Kit from Biosystems (#12812) following the manufacturer’s instructions and using a Benchmark Plus^TM^ spectrophotometer and Microplate Manager^®^ software Version 6.3 (Bio-Rad Inc., Hercules, CA, USA). The H_2_O_2_ content in total lysates of the adipocytes was determined with an Amplex^®^ Red Hydrogen Peroxide/Peroxidase Assay Kit from Invitrogen (Waltham, MA, USA) (#A22188) following the manufacturer’s instructions and using a FLUOstar Optima Fluorometer from BMG LABTECH (Ortenberg, Germany). The rates of lipolysis were determined spectrophotometrically in adipocytes by measuring the glycerol released to the medium over 24 h, from day 6 to 7 of differentiation, in deproteinized culture media, following the Glycerol kinase (#G6278, Sigma-Aldrich)/Glycerol-3-P dehydrogenase (#G3512, Sigma-Aldrich, St. Louis, MO, USA) method, analyzing the NADH concentration at 340 nm [47]. Data are reported relative to the protein content. The lactate concentration released to the medium in the last 24 h of differentiation was measured using an L-Lactate Assay Kit EnzyChrom^TM^ from BioAssay Systems, Hayward, CA, USA (#ECLC-100), following the manufacturer’s instructions. 

### 4.8. Gas Chromatography-Mass Spectrometry (Gc-Ms) Analyses of Polar Metabolites

We measured the relative amount of the polar metabolites citrate, glutamate, cysteine, asparagine, malate, phenylalanine, serine, glutamine, fumarate, proline, succinate, isoleucine, leucine, valine, glycine, alanine, and acid lactic from adipocyte homogenates and conditioned cell media from day 6 to day 7 of differentiation. At day 7, cell media were collected and frozen and adipocytes were harvested by scraping on ice after the addition of 1:1 MeOH:H_2_O and norvaline (Sigma-Aldrich Inc., St., Louis, MO, USA) as internal standard. Next, both adipocyte homogenates and norvaline-spiked culture media were dual-phase extracted using methanol–water/chloroform extraction [48]; the polar fractions obtained were dried under nitrogen flow and then derivatized using 2% (*v*/*v*) methoxyamine (Sigma-Aldrich Inc., St. Louis, MO, USA) in pyridine and MBTSTFA + 1% TBDMCS (Sigma-Aldrich Inc., St. Louis, MO, USA) [49]. Next, 2 µL of all samples were analyzed using an Agilent 7890 GC coupled to an Agilent 5975C MSD single quadrupole with helium as a carrier gas at a flow rate of 1 mL/min in electron impact ionization mode. The oven temperature was programmed as follows: 100 °C for 3 min, then increased at 10 °C/min to 165 °C, followed by a 2.5 °C/min ramp to 225 °C, a 25 °C/min ramp to 265, a 7.5 °C/min ramp to 300 °C, and a final hold time of 5 min. The metabolites presented in the results section were identified following in-house injected standards (Sigma-Aldrich Inc., St. Louis, MO, USA).

### 4.9. RNA Extraction and Quantitative PCR

Total RNA extraction from adipocytes, reverse transcription, and quantitative PCR (qPCR) was performed as previously described [7]. The primer sequences are listed in Appendix A. Gene expression measurements were normalized to the cDNA of the housekeeping gene acidic-ribosomal protein (*Arp*) using the 2^−ΔΔCT^ method. 

### 4.10. Protein Extraction and Western Blotting

Total protein extracts from adipocytes were obtained and immunoblotted following the previously described procedures [7]. ImageJ (NIH) was used to quantify the blots. Relative densitometric arbitrary units (RDAU) were calculated by normalizing data to the loading control GAPDH. Overall contrast and brightness of the Western blot images were adjusted to help clarify data without distorting the image.

### 4.11. Statistical Analysis 

Data are presented as mean ± SEM. Comparisons between two experimental groups were analyzed using Student’s *t*-test. Comparisons among more than two experimental groups were analyzed with a one-way analysis of variance with Tukey’s honest significant difference post hoc test. The *p*-values of significance are indicated in the figure captions. Data were analyzed using GraphPad Prism 6 software (San Diego, CA, USA). Figures were assembled using Adobe illustrator^®^ software (Adobe Systems Inc., San Jose, CA, USA).

## 5. Conclusions

A deficient expression of the adipokine neuregulin 4 in 3T3-L1 adipocytes promotes altered mitochondrial morphology and a decrease in mitochondrial mass, which results in dysfunctional mitochondria, thereby impacting cellular energy metabolism and leading to oxidative stress and inflammation, which in turn trigger insulin resistance.

## Figures and Tables

**Figure 1 ijms-25-11718-f001:**
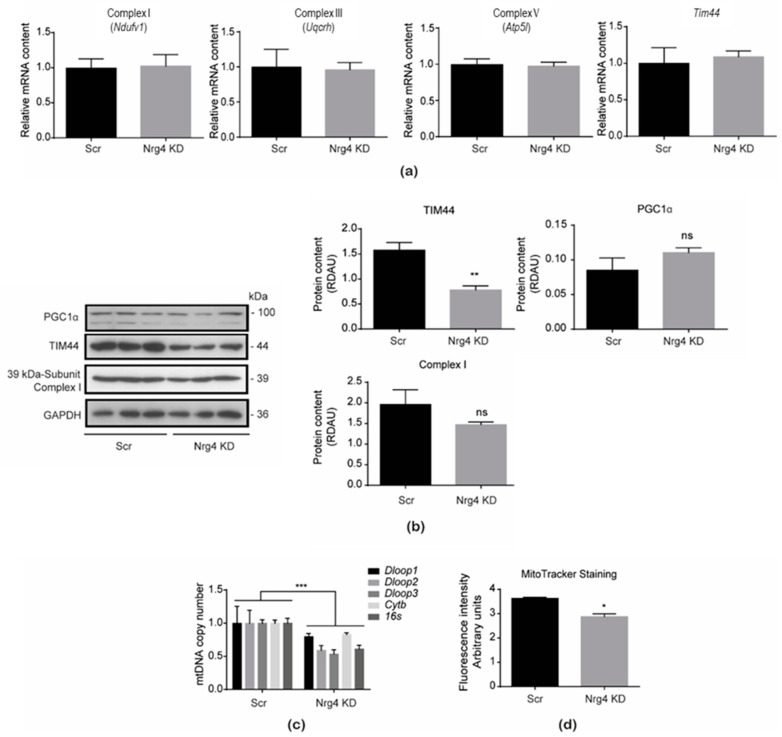
The mitochondrial cellular content is decreased in Nrg4 KD adipocytes. (**a**) Expression of Complex I (*Ndufv1*), Complex III (*Uqcrh*), and Complex V (*Atp5l*) subunits, as well as mitochondrial import inner membrane translocase subunit 44 (*Tim44*) in scrambled (Scr) and Nrg4 knockdown (KD) adipocytes at day 7 (D7) of differentiation (n = 6). (**b**) Representative images and bands’ quantification of the Western blots for the 39 kDa subunit of Complex I, peroxisome proliferator-activated receptor γ co-activator 1-α (PGC1α), and TIM44 in Scr and Nrg4 KD adipocytes at D7. Each lane represents an independent experiment (n = 3). (**c**) Mitochondrial DNA (mtDNA) copy number by qPCR of Scr and Nrg4 KD adipocytes (n = 3). The *Dloop1*, *Dloop2*, *Dloop3*, *cytochrome b* (*Cytb*), and *ribosomal ARN 16S* (*16s*) genomic regions were quantified. (**d**) Fluorometric intensity of MitoTracker^TM^ green FM-stained Scr and Nrg4 KD adipocytes (n = 3) at 575 nm by flow cytometry. Data are mean ± SEM; * *p* < 0.05, ** *p* < 0.01, *** *p* < 0.001, and ns (not significant).

**Figure 2 ijms-25-11718-f002:**
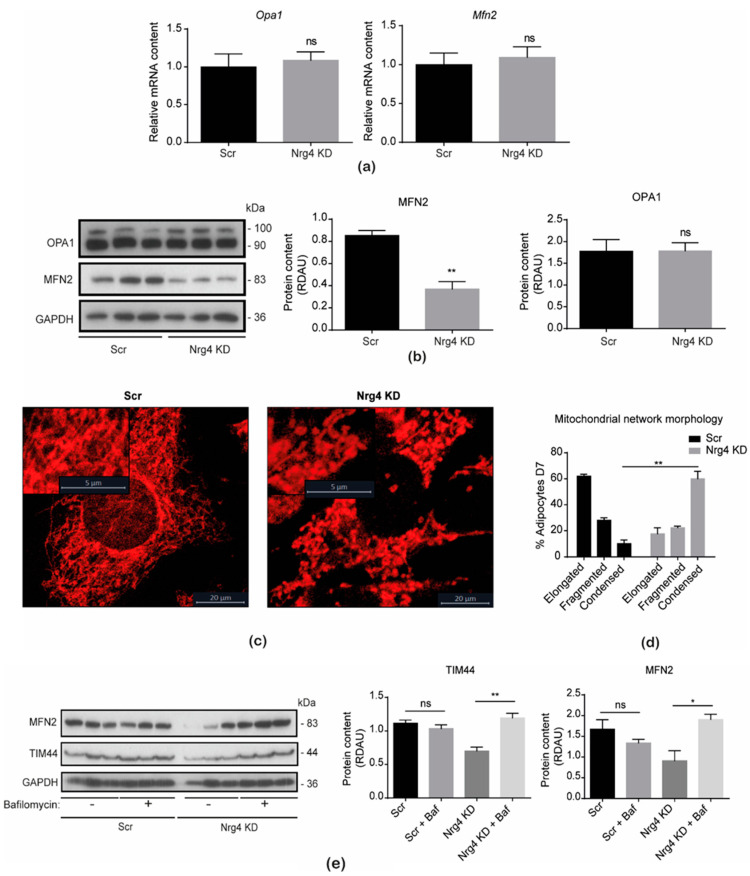
Mitochondrial morphology is altered in Nrg4 KD adipocytes. (**a**) Expression of optic atrophy 1 (*Opa1*) and mitofusin 2 (*Mfn2*) in scrambled (Scr) and Nrg4 knockdown (KD) adipocytes at day 7 (D7) of differentiation (n = 6). QPCR data were normalized to Scr adipocyte expression at D7. (**b**) Quantification and representative Western blot of MFN2 and OPA1 in Scr and Nrg4 KD adipocytes at D7 (n = 3). Each lane represents an independent experiment. (**c**) Mitochondrial micrographs of Scr and Nrg4 KD adipocytes at D7. Images were taken with a Leica TCS-SPE confocal microscopy (×63). Mitochondria were stained with the MitoTracker^TM^ red probe. (**d**) Quantification of adipocytes at D7 with elongated (>4 μm), fragmented (1–4 μm), and condensed (<1 μm) mitochondrial network. The percentage over a total of 10 adipocytes is represented. (**e**) Quantification and representative blot of MFN2 and TIM44 in non-treated and bafilomycin A1 (Baf)-treated Scr and Nrg4 KD adipocytes at D7 (n = 3). Adipocytes were starved for 2 h with EBSS media without serum. Cells were then treated with 200 nM Baf for 2 h. Each lane represents an independent experiment. Data are mean ± SEM; * *p* < 0.05, ** *p* < 0.01, and ns (not significant).

**Figure 3 ijms-25-11718-f003:**
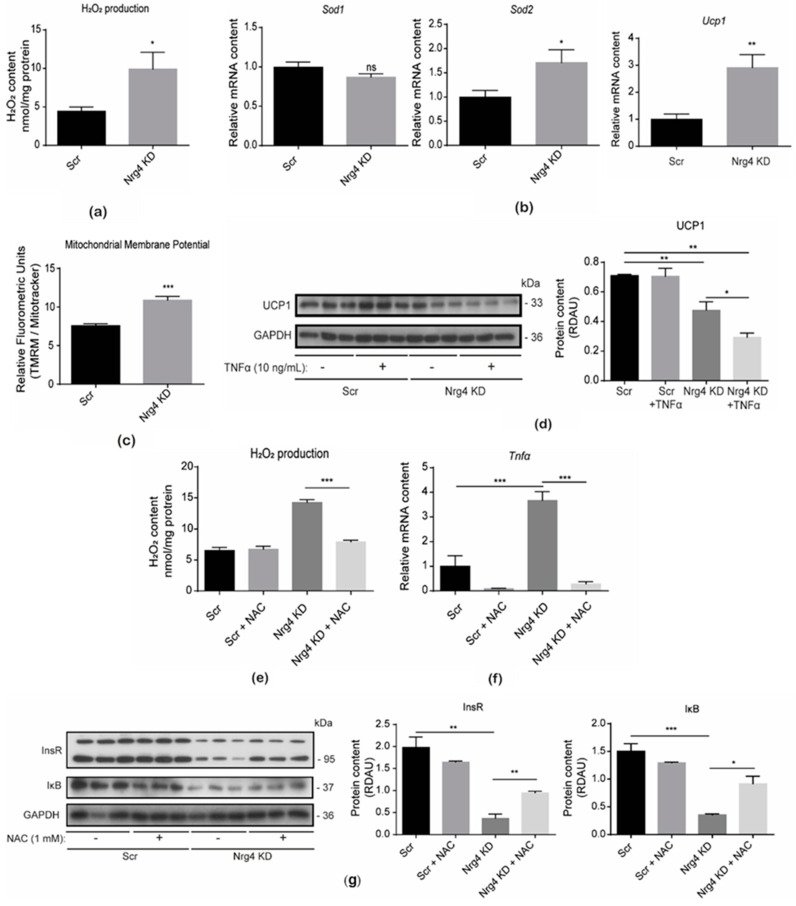
Nrg4 KD adipocytes showed oxidative stress. (**a**) H_2_O_2_ levels in scrambled (Scr) and Nrg4 knockdown (KD) adipocytes at day 7 (D7) of differentiation (n = 4). The H_2_O_2_ levels were detected by fluorometric measurement using the Amplex^®^ Red Assay Kit. (**b**) Expression of superoxide dismutase-1 (*Sod1*), superoxide dismutase-2 (*Sod2*), and uncoupling protein 1 (*Ucp*1) in Scr and Nrg4 KD adipocytes at day 7 (D7) of differentiation (n = 6). Expression was normalized to D7 Scr adipocyte expression. (**c**) The mitochondrial membrane potential of Scr and Nrg4 KD normalized to mitochondrial mass via MitoTracker^®^ green-staining (n = 4). Mitochondrial membrane potential was determined with the MitoProbeTM TMRM staining. The fluorometric intensity was measured by flow cytometry. (**d**) Quantification and representative Western blot of UCP1 in non-treated and TNFα-treated Scr and Nrg4 KD adipocytes at D7 (n = 3). Adipocytes were treated with 10 ng/mL TNFα at day 6 (D6) for 24 h. (**e**) H_2_O_2_ levels in non-treated and N-acetylcysteine (NAC)-treated Scr and Nrg4 KD adipocytes at D7 of differentiation (n = 3). Cells were treated with 1 mM NAC for 24 h. (**f**) Expression of *Tnfa* in non-treated and NAC-treated Scr and Nrg4 KD adipocytes at D7 (n = 3). QPCR data were normalized to non-treated Scr adipocyte expression. (**g**) Quantification and representative Western blot images of the insulin receptor (InsR) and IκB in non-treated and NAC-treated Scr and Nrg4 KD adipocytes at D7 (n = 3). In the Western blots, each lane represents an independent experiment. Data are mean ± SEM; * *p* < 0.05, ** *p* < 0.01, and *** *p* < 0.001 and ns (not significant).

**Figure 4 ijms-25-11718-f004:**
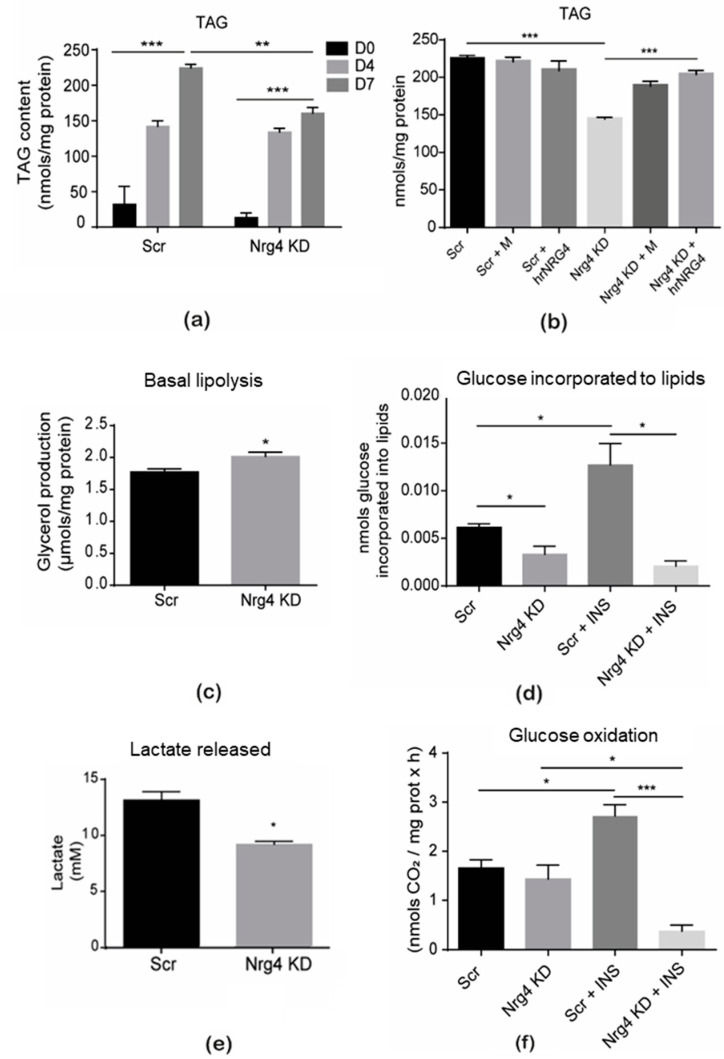
Lipid and glucose metabolism were altered in Nrg4 KD adipocytes. (**a**) Triacylglyceride (TAG) content was measured at day 0 (D0), 4 (D4), and 7 (D7) of differentiation in scrambled (Scr) and Nrg4 knockdown (KD) 3T3-L1 cells (n = 3). (**b**) TAG content in non-treated, Scr-conditioned media (M)-treated, and human recombinant Nrg4 (hrNRG4)-treated adipocytes at D7 (n = 3). TAG content was determined using the commercial kit Enzymatic-Spectrophotometric Glycerol Phosphate Oxidase/Peroxidase from Biosystems. Absorbance was read at 540 nm. Spectrophotometric data were normalized to mg of protein. (**c**) Glycerol released to the media by Scr and Nrg4 KD adipocytes at D7 (n = 3). Glycerol produced is relative to mg of protein of the cell extracts. (**d**) Glucose incorporated into lipids was determined by incubating cells, in 6-well plates, with [^14^C]-glucose for 90 min and measuring radioactivity in the lipidic fraction using a beta-counter. Scr and Nrg4 KD adipocytes, at D7, were studied in non-treated and insulin (INS)-treated conditions. Results indicate the nmol of glucose incorporated into lipids per well (n = 3). (**e**) Lactate release from Scr and Nrg4 KD adipocytes into media (n = 3). Media were obtained from D6 to D7. The release of lactate was determined using the EnzyChrom L-Lactate Assay Kit. Absorbance was measured at 570 nm. (**f**) Glucose oxidation assay in non-treated and insulin (INS)-treated Scr and KD adipocytes at D7 (n = 6). Glucose oxidation was determined by measuring radiolabeled CO_2_ production from [^14^C]-glucose for 3 h. Specifically, nmols of radioactive CO_2_ were determined and normalized to mg of protein per hour. Adipocytes were treated with 100 nM insulin for 30 min at 37 °C. Data are mean ± SEM; * *p* < 0.05, ** *p* < 0.01, and *** *p* < 0.001.

**Figure 5 ijms-25-11718-f005:**
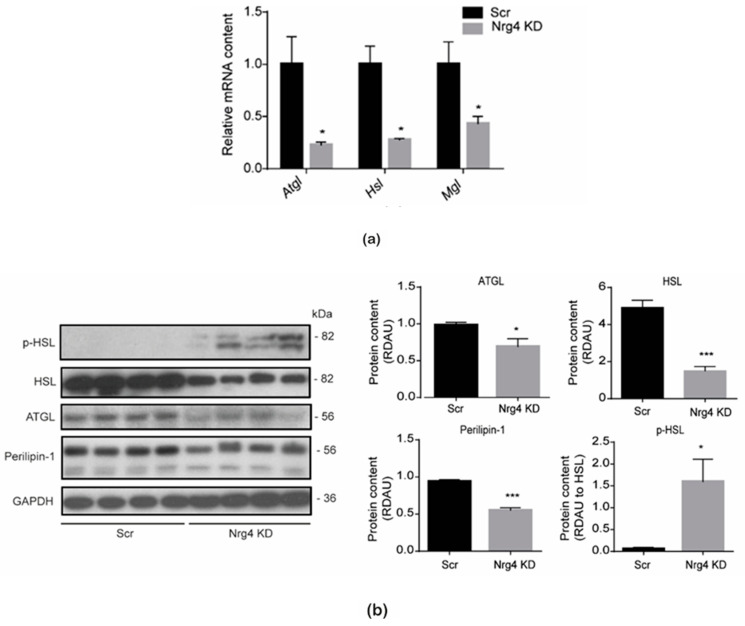
Classical lipolysis in Nrg4 KD adipocytes. (**a**) Expression of adipose triacylglyceride lipase (*Atgl*), hormone-sensitive lipase (*Hsl*), and monoacylglycerol lipase (*Mgl*) in scrambled (Scr) and Nrg4 knockdown (KD) adipocytes at day 7 (D7) of differentiation (n = 3). Data were normalized to Scr adipocyte expression at D7. (**b**) Quantification and representative Western blot of total ATGL, total HSL, and perilipin-1, as well as phospho (p)-HSL in Scr and Nrg4 KD adipocytes at D7 (n = 4). Each lane represents an independent experiment. Data are mean ± SEM; * *p* < 0.05, and ****p* < 0.001.

**Figure 6 ijms-25-11718-f006:**
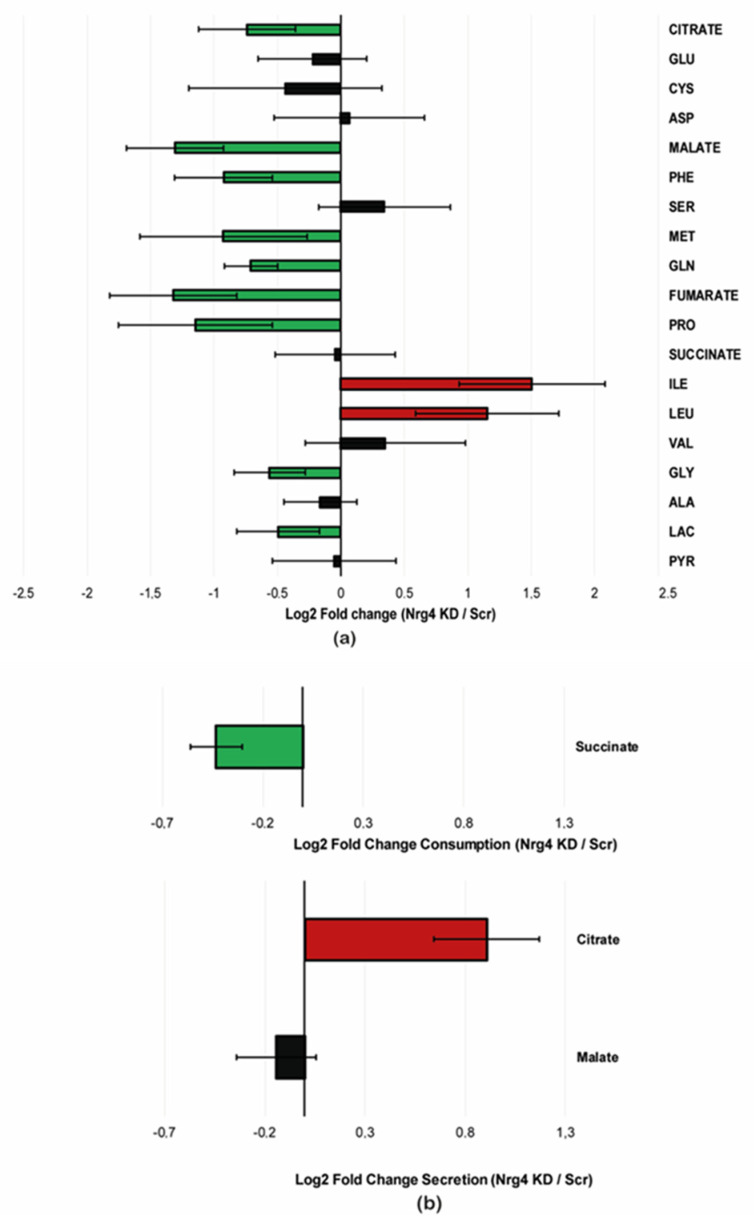
The relative abundance of polar metabolites in scrambled and Nrg4 KD adipocytes. (**a**) Intracellular content of the following metabolites: citrate, glutamate (GLU), cysteine (CYS), asparagine (ASP), malate, phenylalanine (PHE), serine (SER), glutamine (GLN), fumarate, proline (PRO), succinate, isoleucine (ILE), leucine (LEU), valine (VAL), glycine (GLY), alanine (ALA), lactic acid (LAC), and pyruvate (PYR) in Scr and Nrg4 KD adipocytes at day 7 (D7). Data were obtained from GC-MS analysis of cellular lysates (n = 4). Data are represented in Log2 fold change (KD/Scr). (**b**) Production/consumption rates of succinate, citrate, and malate in Nrg4 KD and Scr adipocytes. Data were obtained from GC-MS analysis of conditioned media from D6 to D7. Data are represented in Log2 fold change [(D7 Nrg4 KD-D6 Nrg4 KD)/(D7 Scr-D6 Scr)]. In green, the values are significantly lower in Nrg4 KD adipocytes; in red, the values are significantly higher in Nrg4 KD adipocytes; in black, significant changes were not reached. Data are mean ± SD.

## Data Availability

Generated data are available upon request.

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
