# Peer review of "Neuregulin 4 Downregulation Alters Mitochondrial Morphology and Induces Oxidative Stress in 3T3-L1 Adipocytes"

_ijms, 2024, doi:10.3390/ijms252111718_

Round 1

Reviewer 1 Report

Comments and Suggestions for Authors

Comments to the authors:

Overall, this is a well-executed study with the potential for significant impact in the field of metabolic diseases and mitochondrial research. I have some minor concerns that need to be addressed by the authors before publication.

1: In Figure 1b, why the complex V is much different in Scr group. Looks like one sample did not express complex V.  Further, the authors mentioned adiposities silenced for the expression of the adipokine neuregulin showed a lower mitochondrial content. It would be better to show the staining data in Figure 1d. Also, the Nrg4 expression level should be shown by Western blotting.

2: In Figure 2, the Nrg4 knockdown should be shown. The figure 2c is not convincing. The mitochondrial marker protein should be shown, such as Tom20 to confirm the hypothesis.

3: In Figure 3, the ROS level should be shown and why the total InsR and IkB expression level was reduced. What is the mechanism?

4: In Figure 5, why the total HSL is was reduced in Nrg4-KD group? How does the Nrg4 regulate HSL expression?

5: It is a good practice to see the mitochondrial morphology and related markers in iWAT of Nrg4-KO mice.

Comments on the Quality of English Language

The English could be improved to express the research more clearly.

Author Response

Review - 1

Comments to the authors:

Overall, this is a well-executed study with the potential for significant impact in the field of metabolic diseases and mitochondrial research. I have some minor concerns that need to be addressed by the authors before publication.

Response to the Reviewer - 1

Many thanks for your comments. Here we answer your questions:

Point 1. In Figure 1b, why the complex V is much different in Scr group. Looks like one sample did not express complex V.  Further, the authors mentioned adiposities silenced for the expression of the adipokine neuregulin showed a lower mitochondrial content. It would be better to show the staining data in Figure 1d. Also, the Nrg4 expression level should be shown by Western blotting.

Response to Point 1:

  • Regarding Complex V - In the Western blot assay for the a subunit of the Complex V, there was a band detection interference. We wanted to faithfully preserve the original images and show them as they were obtained despite observing this defect. Since the reviewer might consider that the inclusion of this data may not be appropriate, we have eliminated it in the revised version of the manuscript (Results, section 2.1 and Figure 1b).
  • Mitochondria content – In the figure 1 we developed the concept of cellular mitochondrial content, whereas in the figure 2 we analyzed the mitochondrial dynamics. Figure 2c, showing mitochondria staining, does not give information on the total mitochondrial cell content, what it shows is the mitochondrial morphology disarrangement, regardless of the total amount of mitochondria.
  • Nrg4 western blotting – There was no available antibody for neuregulin 4 reliable for western blot assays when we did these studies. Just recently we have assayed a promising antibody for neuregulin 4, according to Shi et al. Nature Metab. 2022. No band was obtained using positive controls. In contact with the commercial providers and reporting the results of our assays, there was concern on the quality of this antibody. We are currently assaying another antibodies provided by the same company and in contact with the technical services. No other article has confirmed published data with the initially assayed antibody. So, we are dealing with this matter and this is the reason that, so far, we cannot give data of neuregulin 4 protein content.

Point 2. In Figure 2, the Nrg4 knockdown should be shown. The figure 2c is not convincing. The mitochondrial marker protein should be shown, such as Tom20 to confirm the hypothesis.

Response to Point 2:

  • Nrg4 knockdown should be shown - The expression of the Nrg4 in the stable cell line Nrg4 KD is shown in the supplementary figure 1, in comparison with the expression observed in the scramble cell line. Since a detailed description of this cell line was done in a previous article (Diaz-Saez, Int J Mol Sci, 2021, included in the references), here we only provide the qPCR results for Nrg4 at the time point in which these experiments were done in order to prove that the Nrg4 KD cell line was preserving the silencing of the Nrg4 gene expression.
  • Figure 2c is not convincing – We understand the concern of the reviewer just analyzing one image of each cell line. For this reason, we provided a quantification of the mitochondrial elongation (Fig. 2d) based on standard criteria that is, considering a range of x<1 mm to x>4 mm to distinguish among elongated, fragmented and condensed mitochondrial network. Data was obtained from 10 adipocytes from each cell line. In this sense, there is extensive bibliography from Zorzano’s laboratory that supports this procedure (e.g. Sebastian, PNAS, 2012; Muñoz, EMBO J, 2013, included in the references).
  • mitochondrial marker protein should be shown, such as Tom20 to confirm the hypothesis – We initially analyzed the expression of two mitochondrial markers, TIM44 (shown in Figure 1) and TOM20 (not shown in the manuscript). Here we add the data obtained by qPCR for TOM20 (n = 6) for the reviewer:

No significant differences were obtained in the mRNA levels for TIM44 (manuscript) nor for TOM20. It should be noted that TIM44 is not involved in mitochondrial dynamics, it is a subunit of a translocase of the mitochondrial inner membrane, and it is used as a mitochondrial marker. Therefore, no further studies were performed to analyze TOM20 at the protein level, since TIM44 provided this information.

Point 3. In Figure 3, the ROS level should be shown and why the total InsR and IkB expression level was reduced. What is the mechanism?

Response to Point 3:

  • ROS level should be shown - In figure 3a, we show the ROS levels, specifically we measured the cell content of H2O2 by fluorimetry using the Amplex® Red Assay Kit (described in the methods and indicated in the figure 3 legend).
  • why the total InsR and IkB expression level was reduced. What is the mechanism? - The mentioned data (manuscript, Figure 3g) are in accordance with a previous article from our laboratory (Díaz-Sáez, 2021, figures 3 and 5), included in the manuscript submission to facilitate the task of the reviewers. There, we described that Nrg4 KD adipocytes show an autonomous inflammation profile characterized by a high expression of proinflammatory cytokines, a reduced expression of the insulin receptor and of IkB, which prevents the action of the transcription factor NFkB involved, in turn, in the expression of the proinflammatory cytokines. We described that the treatment of Nrg4 KD adipocytes with an anti-inflammatory agent, sodium salicylate, allowed the recovery of the insulin receptor content as well as the IkB content, while the levels of proinflammatory cytokines decreased. The negative effects of the proinflammatory cytokines on the insulin receptor expression had been previously described and reported in the Discussion of the aforementioned article. Thus, the autonomous inflammation displayed by Nrg4 KD adipocytes is responsible for the loss of the insulin receptor and the inhibitor of NFkB (IkB).

Point 4. In Figure 5, why the total HSL is was reduced in Nrg4-KD group? How does the Nrg4 regulate HSL expression?

Response to point 4:

  • The most probable responsible is, again, TNFa since its effect inhibiting HSL gene expression has been previously reported in 3T3-L1 adipocytes (Sumida et al. J. Biochem. 107, 1-2, 1990). Since we and others reported that Nrg4 is an anti-inflammatory adipokine, the effect observed on HSL in Nrg4 KD adipocytes is expected to be a consequence of the reduced level of Nrg4 and the raise in the autonomous inflammation. In this sense, here we join unpublished data, for the reviewer, showing that the treatment with the glucocorticoid dexamethasone (D, 200 nM, for the last 48 hours of the adipocytes differentiation, from day 5 to day 7) allows restoring the HSL expression. So again, the autonomous inflammation enforced by the reduced expression of Nrg4 in KD adipocytes, looks to be the cause of the reduced HSL expression.

Point 5. It is a good practice to see the mitochondrial morphology and related markers in iWAT of Nrg4-KO mice.

Response to point 5:

  • The study of the mitochondrial morphology in adipocytes isolated from iWAT is highly difficult due to the large size of the lipid droplet that enforce mitochondria to be quartered at the subsarcolemmal area. So far, up to our knowledge, there is no published data in this regard. Studies on the mitochondria dynamics and morphology have been done mostly in cultured cell lines.

Comments on the Quality of English Language

The English could be improved to express the research more clearly.

Response to the Quality of English:  The manuscript has been submitted to an English Language revision and the changes are included in the reviewed version.

Submission Date

28 September 2024

Date of this review

06 Oct 2024 03:21:35

Date of response to Reviewer:

23/10/2024

Reviewer 2 Report

Comments and Suggestions for Authors

The current study is an important piece of work in identifying the potential drivers of Nrg4.  A comprehensive invitro study involving flow cytometry, confocal microscopy, quantifying the mitochondrial DNA, measured glucose oxidation and lipogenesis at different time points, and GC-MS to measure polar metabolites in between SCN and Nrg4 KD cells was conducted. This study is an increment to their previous study.

The material and method was well written. 

The authors have done all the necessary experiments to show the potential drivers that are associated with insulin resistance in Nrg4.

The results are appropriately presented.

The discussion is well written.

Overall the study is complete.

However, there are minor comments to make:

1) FIG 1A Change the style from normal to italic Tim44.

The author needs to look for changes in style at several places in the manuscript.

2) Minor spelling mistake confirmed in place of Conformed

Author Response

Review - 2

Comments and Suggestions for Authors

The current study is an important piece of work in identifying the potential drivers of Nrg4.  A comprehensive invitro study involving flow cytometry, confocal microscopy, quantifying the mitochondrial DNA, measured glucose oxidation and lipogenesis at different time points, and GC-MS to measure polar metabolites in between SCN and Nrg4 KD cells was conducted. This study is an increment to their previous study.

The material and method was well written.

The authors have done all the necessary experiments to show the potential drivers that are associated with insulin resistance in Nrg4.

The results are appropriately presented.

The discussion is well written.

Overall, the study is complete.

However, there are minor comments to make:

Response to the Reviewer - 2

Many thanks for your interest in our work. Here we answer your questions:

Point 1. FIG 1A Change the style from normal to italic Tim44. The author needs to look for changes in style at several places in the manuscript.

Response to Point 1:

  • In figure 1a, Tim44 was written in italic style, but certainly, the subunits of the Complex I, III and V should be indicated in italic style. This information has been included in the reviewed version of the manuscript.
  • The whole manuscript has been reviewed for the style and characters used, when mentioning genes expression or proteins. Thanks for your suggestion.

Point 2. Minor spelling mistake confirmed in place of Conformed

Response to Point 2:

  • Manuscript was submitted to an English language review and CONFORMED was changed by FORMED, in reference to the formation of mitochondrial condensed structures (line 129 of the reviewed manuscript).

Date of response to Reviewer: 23/10/2024

Round 2

Reviewer 1 Report

Comments and Suggestions for Authors

The authors have addressed all the concerns

Comments on the Quality of English Language

It is fine